# Effect of the Heating Rate to Prevent the Generation of Iron Oxides during the Hydrothermal Synthesis of LiFePO_4_

**DOI:** 10.3390/nano11092412

**Published:** 2021-09-16

**Authors:** Francisco Ruiz-Jorge, Almudena Benítez, M. Belén García-Jarana, Jezabel Sánchez-Oneto, Juan R. Portela, Enrique J. Martínez de la Ossa

**Affiliations:** 1Department of Chemical Engineering and Food Technology, Faculty of Sciences, International Excellence Agrifood Campus (CeiA3), University of Cadiz, 11510 Puerto Real, Spain; franciscojavier.ruiz@uca.es (F.R.-J.); belen.garcia@uca.es (M.B.G.-J.); jezabel.sanchez@uca.es (J.S.-O.); enrique.martinezdelaossa@uca.es (E.J.M.d.l.O.); 2Department of Inorganic Chemistry and Chemical Engineering, University Institute of Nanochemistry (IUNAN), University of Cordoba, 14071 Córdoba, Spain; q62betoa@uco.es

**Keywords:** lithium iron phosphate, hydrothermal synthesis, heating rate, morphology, crystallinity and purity

## Abstract

Lithium-ion batteries (LIBs) have gained much interest in recent years because of the increasing energy demand and the relentless progression of climate change. About 30% of the manufacturing cost for LIBs is spent on cathode materials, and its level of development is lower than the negative electrode, separator diaphragm and electrolyte, therefore becoming the “controlling step”. Numerous cathodic materials have been employed, LiFePO_4_ being the most relevant one mainly because of its excellent performance, as well as its rated capacity (170 mA·h·g^−1^) and practical operating voltage (3.5 V vs. Li^+^/Li). Nevertheless, producing micro and nanoparticles with high purity levels, avoiding the formation of iron oxides, and reducing the operating cost are still some of the aspects still to be improved. In this work, we have applied two heating rates (slow and fast) to the same hydrothermal synthesis process with the main objective of obtaining, without any reducing agents, the purest possible LiFePO_4_ in the shortest time and with the lowest proportion of magnetite impurities. The reagents initially used were: FeSO_4_, H_3_PO_4_, and LiOH, and a crucial phenomenon has been observed in the temperature range between 130 and 150 °C, being verified with various techniques such as XRD and SEM.

## 1. Introduction

The global and increasing energy demand, and the need to replace the consequential consumption of fossil fuels because of environmental concerns, has generated a growing interest, not only in the development of renewable sources of energy, but also in the design of more advanced energy storage systems such as lithium-ion batteries (LIBs) [1], super capacitors [2], lithium sulfur batteries [3], sodium sulfur batteries [4], and redox flow batteries [5] with improved energy density and cycling performance. Nowadays, LIBs are important energy storage devices because of their high specific energy, low self-discharge, excellent cycle performance, no memory effect, and lesser environmental impact [6]. About 30% of the manufacturing budget for LIBs is spent on cathode materials, and its level of development is lower than that of the negative electrode, the diaphragm, or the electrolyte. Therefore, it is the “control step” that determines the battery performance in terms of working voltage, energy density, and rate performance [6]. Numerous cathodic materials have been employed, such as LiMn_2_O, Li_3_V_2_(PO_4_)_3_, and LiCoO_2_, but it is LiFePO_4_ that has become the main one used because of its excellent performance, as well as its high theoretical specific capacity (170 mA·h·g^−1^), practical operating voltage (3.5 V vs. Li^+^/Li), long life cycle, superior safety, low cost, low toxicity, abundant resources, and lesser environmental impact [7].

To date, many studies have focused on the production of LiFePO_4_ particles by different methods such as: solid state synthesis [8,9,10], mechanochemical activation [10,11], sol−gel synthesis [10,12], coprecipitation [10,13,14], and hydrothermal synthesis [6,10], among others [7]. Among all of them, hydrothermal synthesis has been gaining prominence since its capacity to produce extremely pure and crystalline particles using relatively low temperatures (115–400 °C) [15,16] has been demonstrated, and in relatively short reaction times (from seconds to hours) [17,18].

For a long time, the research works that have been conducted on hydrothermal synthesis have primarily focused on the improvement of the two main drawbacks that LiFePO_4_ presents. These are its low diffusivity (10–14 cm^2^·s^−1^) and its low electrical conductivity (10^−9^ S·cm−^1^) [1]. Several studies have tried to find the way to improve lithium ion diffusivity by varying the size and morphology of its particles through the control of the different variables that have an influence on their formation process in order to shorten lithium-ion main diffusion channel ([10] channel): pH [19,20], reaction time [15,21], stirring [22], temperature [16,23], and use of surface-modifying agents and reducing agents [24,25,26,27,28]. Other studies have tried to improve lithium-ion electrical conductivity by coating its particles with carbon or doping them with transition metals [29,30,31].

In addition, another important part of the studies on hydrothermal synthesis have been based on the rapid and economic production of LiFePO_4_ [17,32]. To accomplish an efficient production of LiFePO_4_, continuous processes, where preheating is usually rapid, are required. However, the use of high heating rates may generate large amounts of iron oxide impurities (magnetite) and poor particle crystallization, which prevents good electrochemical performance [33,34]. In our previous work [34], we found that magnetite was generated at heating rates of 86 °C/min. This did not occur when heating rates of 5.26 °C/min were used. In addition, it was noted that the particles were less crystalline when rates of 86 °C/min were applied. One alternative approach to avoid magnetite formation consists of the use of reducing agents such as ascorbic acid, pyrrole, urea, sucrose, citric acid, etc. [25,26,35,36] which might increase the cost of the process. The main objective of this work is to reduce the overall process reaction time while obtaining a LiFePO_4_ as pure as possible, i.e., with less magnetite impurities and without any use of reducing agents. For this purpose, we have focused on the implementation of various combinations of slow and fast heating rates in order to determine the optimal time to start the fast heating step in a process that would start with a slow heating rate. Different hydrothermal synthesis experiments where a slow heating rate from room temperature to 50, 100, 150, 150, 200, 250, and 300 °C was followed by a fast heating stage to reach 300 °C have been carried out. A particularly crucial phenomenon has been observed in the temperature range between 130 and 150 °C, where slow heating rates produce particles with less magnetite impurities and higher crystallinity.

## 2. Materials and Methods

### 2.1. Equipment and Experimental Procedures

As a first stage of the work, crystalline LiFePO_4_ particles were synthesized in a 0.3 L commercially available stainless-steel reactor (bolted closure packless autoclave) described in detail in a previous work [37]. These LiFePO_4_ particles would be used as a reference or base to be compared with the particles obtained, in the second part of the study (14 mL batch reactors), using the combination of slow and fast heating rates, under the same temperature conditions. The exact moment at which the first crystalline nuclei of LiFePO_4_ were generated in the reactor was also determined, with the aim of clarifying and solidly supporting the results obtained. The reactors fill ratio was adjusted so that the heating times were as close as possible between the two reactors in order to extrapolate the data.

The procedure to generate the base LiFePO_4_ consisted of first placing 2.0392 g of LiOH·H_2_O and 4.5039 g of FeSO_4_·7H_2_O in the reactor. Then, 180 mL of deionized water and 1.248 mL of H_3_PO_4_ were added, the reactor was closed, and the mixture was stirred and purged with nitrogen to remove the oxygen. The selected molar ratio, in order to adjust the pH of mixture at 7, has been the same as the one used for the generation of LiFePO_4_ particles using different configurations of heating rates in this work, being optimized and described in detail in a previous work [34] (Fe: PO_4_: Li; 0.9: 1.2: 2.7). The reaction medium was then heated up to 300 °C at a rate of 4 °C/min and cooled down to room temperature without stirring. Upon completion of the hydrothermal synthesis experiments, the final solution was collected from the reactor, centrifuged to separate the LiFePO_4_ particles located at the bottom of the supernatant, and dried by means of an oven at 80 °C. More specific details can be found in our previous work [34].

For the study on the formation of the first crystalline nuclei of LiFePO_4_, the same procedure was used as for the generation of base LiFePO_4_. In this case, 3 experiments were carried out, where temperature was raised up to 130, 140, and 150 °C, respectively, at a rate of 4°C/min, and then cooled down to room temperature.

For the generation of LiFePO_4_ particles using different configurations of slow and fast heating rates, two heating systems were combined. The first method, consisting of a sand bath (Techne model TC-8D with a power of 4 kW), had been used in our previous work [34]. For the present work, a new method with a fluidized sand bath (Techne model SBL-2 with a power of 3 kW) was added. Thus, the heating system was made up of two fluidized sand baths, two compressors that introduced air into the baths to fluidize the sand, a temperature controller, and a reactor manufactured by our research team as detailed in a previous work [34]. The experimental procedure used to fill the 14 mL volume batch reactors to perform the hydrothermal synthesis of LiFePO_4_ from the selected reagents, and the separation of the corresponding LiFePO_4_ microcrystals obtained had been optimized and described in detail in a previous work [34]. The procedure is highly similar to that of the base LiFePO_4_, only that to enable agitation in small reactors, the necessary water is added in two parts.

### 2.2. Reagents, Process Reactions and Products

The LiFePO_4_ was generated by means of the following precursors: LiOH·H_2_O (99%), FeSO_4_·7H_2_O (98%), H_3_PO_4_ (85%), and deionized water. All the chemicals were supplied by Aldrich Co.

The process to generate the LiFePO_4_, could be summarized in two stages. The first and shorter stage consists of the dissolution of the initial reagents (FeSO_4_ + H_3_PO_4_ + 3LiOH) in the water, which results in the formation of the intermediated reagents (Li_3_PO_4_ and Fe_3_(PO_4_)_2_·8H_2_O) [38]. During the second stage, the intermediate reagents, first Li_3_PO_4_ and finally Fe_3_(PO_4_)_2_·8H_2_O, are dissolved to yield LiFePO_4_ and Li_2_SO_4_ [33]. The LiFePO_4_ particles are generated when the Fe_3_(PO_4_)_2_·8H_2_O liberates Fe^+2^ into the medium. The global equation for LiFePO_4_ formation could be expressed as follows:FeSO_4_ + H_3_PO_4_ + 3LiOH → LiFePO_4_ + Li_2_SO_4_ + 3H_2_O

When the heating rate of the reactors during the formation of LiFePO_4_ is extremely high, large concentrations of Fe^+2^ are quickly released into the medium, which favors the generation of magnetite (Fe_3_O_4_) because of the Schikorr reaction [39].

Schikorr reaction:2 (Fe^+2^ → Fe^+3^ + e^−^)(1)
2 (H_2_O + e^−^ → ½ H_2_ + OH^−^)(2)
2 Fe^+2^ + 2 H_2_O → 2 Fe^+3^ + H_2_ + 2 OH^−^(3)

The presence of the Fe^+2^ leads to:3 Fe^+2^ + 2 H_2_O → Fe^+2^ + 2 Fe^+3^ + H_2_ + 2 OH^−^(4)

Electroneutrality requires the iron cations on both sides of the equation to be counterbalanced by 6 hydroxyl anions (OH^−^):3 Fe^+2^ + 6 OH^−^ + 2 H_2_O → Fe^+2^ + 2 Fe^+3^ + H_2_ + 8 OH^−^(5)
3 Fe (OH)_2_ + 2 H_2_O → Fe (OH)_2_ + 2 Fe (OH)_3_ + H_2_(6)

Due to the autoproteolysis of the hydroxyl anions:OH^−^ + OH^−^ → O^−2^ + H_2_O(7)

Therefore:3 Fe (OH)_2_ + 2 H_2_O → (FeO + 1 H_2_O) + (Fe_2_O_3_ + 3 H_2_O) + H_2_(8)
3 Fe (OH)_2_ → FeO + Fe_2_O_3_ + 2 H_2_O + H_2_(9)

Finally, following the magnetite formation the following reaction takes place:FeO + Fe_2_O_3_ → Fe_3_O_4_(10)

Therefore, by increasing the Fe^+2^ concentration in the medium, the Schikorr reaction might accelerate.

In order to evaluate the possibility of improving the overall process, this study has focused in detailed the effect caused by different heating configurations, so that increments in the heating rate can be applied at different times during the hydrothermal synthesis. Thus, this influence from the different heating configurations has been closely analyzed to determine the critical temperature levels during the heating stage where a high heating rate can be implemented to improve the LiFePO_4_ synthesis process, not only in terms of the crystallinity, morphology, and purity of the particles, but also with regard to the feasibility of the process.

### 2.3. Formation of the Base LiFePO_4_

In order to carry out an adequate comparison of the crystallinity, purity, and morphology of the particles obtained from the subsequent experiments, we previously produced our own LiFePO_4_ particles to be used as a reference. These will be referred to here as the “base LiFePO_4_”. Such base LiFePO_4_ particles have to be generated under tightly controlled temperature and heating rate conditions (See Figure 1) according to the literature [15,16,18,23,24,40]. The reactor and experimental procedure have been described in Section 2.1. The base experiment was carried out under the same conditions as the rest of the experiments conducted in this work: i.e., the reaction medium was heated up to 300 °C, reached a pressure of 103 bar, was held under such conditions for 5 min, and then slowly cooled down to room temperature.

### 2.4. Study of the Formation of the First Crystalline Nuclei of LiFePO_4_

In order to perform an accurate analysis of the results from this study, it is necessary to determine the exact moment when the first crystalline nuclei of LiFePO_4_ appear. Given that LiFePO_4_ is generated when Fe_3_(PO_4_)_2_-8H_2_O begins to dissolve, and that this depends on the length of time required to heat the reaction medium, such length of time required by our reactors should be precisely determined in order to clarify this key point regarding the purity and crystallization of LiFePO_4_. According to the study carried out by J. Lee and A.S. Teja [16], the formation of the first crystalline nuclei would take place when the medium reaches between 120 and 190 °C, since only Fe_3_(PO_4_)_2_-8H_2_O appeared in the samples obtained at 120 °C, while the LiFePO_4_ particles were already visible in the samples produced at 190 °C. On the other hand, C. Min et al. [40], established that a large amount of LiFePO_4_ nuclei are rapidly formed when the medium temperature reaches values around 124–130 °C.

Considering the temperatures that have been already studied in the bibliography, the experiments were conducted at 130, 140, and 150 °C (reaching 1.01, 6.55 and 7.58 bar, respectively). The reactor and the experimental procedures have been described in Section 2.1.

### 2.5. Formation and Separation of LiFePO_4_ Microcrystals

The reactor heating procedure was as follows: First, the reactor was submerged into one of the fluidized sand baths still at room temperature (the bath heating system had not been turned on yet). Once inside, the sand bath was turned on and a slow heating rate of 4.44 °C/min was applied. When the target temperature set for each experiment was reached, the reactor was submerged into another sand bath, which had been previously heated and maintained at a constant temperature of 300 °C. Therefore, from the moment the reactor was submerged into the second sand bath until the moment the reactor reached 300 °C, the medium was subjected to a fast heating rate. Thereafter, the reactor was maintained at 300 °C for 5 min (reaction time) before being removed from the sand bath and kept in contact with the laboratory air until completely cooled down to room temperature (see Figure 2). Thus, the final reaction temperature was the same for all the experiments, but the heating time, and therefore the total time of each experiment, which comprises the heating time plus the 5 min reaction time at 300 °C, was different.

Table 1 includes the operating conditions of the different hydrothermal synthesis (HS) experiments that combine slow and fast heating. In addition, in order to further consider the configuration range to be tested, two experiments with only one type of heating were carried out: (i) thus, experiment F300 was only subjected to fast heating up to 300 °C, while (ii) experiment S300 was only subjected to slow heating up to the same temperature. Figure 2 shows the evolution of the temperature profiles of the reaction medium over time for the HS experiments included in Table 1.

Once the hydrothermal synthesis experiments were completed, the final solution was collected from the reactor, centrifuged, and dried, following the procedure already explained in Section 2.1.

### 2.6. Characterization

The structural properties of LiFePO_4_ particles were analyzed by X-ray diffraction (XRD) by means of a Bruker D8 Discover A25 diffractometer (Bruker Española S.A., Madrid, Spain) using Cu Kα radiation, Ge monochromator, and a Lynxeye detector. The patterns were registered within the 10−80° (2θ) range, according to a 140 s. step time. The lattice cell parameters, the crystallite size and the amount of impurities present in the synthesized LiFePO_4_ were calculated using Topas software (Bruker Española S.A., Madrid, Spain) according to the full pattern matching method. The morphological properties of the samples were determined by means of a field-emission scanning electron microscope (FESEM) using a FEI-Nova Nano SEM 450 instrument (Izasa Scientific, Madrid, Spain). The presence of magnetite impurities in the synthesized LiFePO_4_ was verified by X-ray diffraction and also by our own-built electromagnet to confirm the magnetic behavior of the samples.

## 3. Results and Discussion

### 3.1. Analysis of the Synthesized Base LiFePO_4_

The purity and crystallinity of the synthesized base LiFePO_4_ was determined by XRD. Figure 3 shows the XRD pattern of the LiFePO_4_ powders synthesized in the HS base experiment, where the peaks in the different diffractograms closely match the standard LiFePO_4_ pattern (JCPDS card no. PDF 40-1267). Moreover, no impurities (no magnetite) were detected in the diffractogram. 

The refinement of the lattice parameters for the orthorhombic structure of the base LiFePO_4_ with the *Pnma* space group provided the following values: a = 10.3328 Å, b = 6.0043 Å, and c = 4.6977 Å, which is in accordance with the literature [41].

The crystallinity of the particles obtained in the base experiment was considered optimum, and, therefore, they were suitable to be used as the reference for this study. That is, the particles obtained from the experiments, where different combinations of slow heating and fast heating was implemented, were to be mainly compared against the crystallinity of the base LiFePO_4._

Figure 4 displays SEM images of the base LiFePO_4_, where flat and well-defined microparticles can be observed. Its morphology is dominated by face (010) that corresponds to diamond-shaped crystallites [42]. Thus, diamond-shaped platelets are less than 1 micron thick and their dimensions vary between 2–4 microns for the shortest diagonal and 6–8 microns for the longest one.

### 3.2. Study on the Formation of the First Crystalline Nuclei of LiFePO_4_

Figure 5 shows the XRD pattern of synthesized LiFePO_4_ powders produced at different temperatures: 130, 140, and 150 °C. As can be observed, the samples that had been synthesized at 140 and 150 °C present the characteristic peaks that can be attributed to the orthorhombic olivine structure of LiFePO_4_. However, the sample that had been synthesized at 130 °C presented no evidence of any LiFePO_4_ content. Although, as previously mentioned, Min et al. [34] established that a large amount of LiFePO_4_ nuclei were formed in the 124–130 °C range. Fe_3_(PO_4_)_2_-8H_2_O may be soluble within that temperature range albeit the heating time employed in this study may have not been long enough to allow the generation of the first crystalline nuclei of LiFePO_4_. Therefore, even though it cannot be affirmed that the first LiFePO_4_ crystalline nuclei are necessarily formed between 124 and 130 °C, it can be presumed that Fe_3_(PO_4_)_2_-8H_2_O begins to dissolve at around 130 °C, so that this temperature level might be key to explain much of what happens in this process, which would make it worth a deeper discussion, as can be seen in Section 3.3.

### 3.3. Study on the Combination of Slow-Fast Heating Rates during the HS of LiFePO_4_ Particles

#### 3.3.1. Crystallinity and Purity of the LiFePO_4_ Particles

The purity and crystallinity of the synthesized particles was determined by XRD. Figure 6 shows the XRD patterns of LiFePO_4_ powders generated through the different experiments included in Table 1 and that corresponding to the base HS experiment. The peaks in all of the diffractograms closely match the reference pattern (JCPDS card no. PDF 40-1267). Therefore, the hydrothermal synthesis of LiFePO_4_ particles has been confirmed for all the experiments that have been carried out. On the other hand, the XRD patterns revealed magnetite impurities in some of the samples, as can be seen in Figure 6.

Table 2 shows that only pure LiFePO_4_ particles were generated in experiments S300 and S250F. On the other hand, all the particles that had been obtained from the experiments that implemented a fast heating rate before reaching 250 °C (F300, S50F, S100F, S150F, and S200F) presented magnetite impurities to a greater or lesser degree. This could be due to the dissolved form of Fe_3_(PO_4_)_2_·8H_2_O. It has been already mentioned (Section 2.2) that LiFePO_4_ is generated when Fe_3_(PO_4_)_2_·8H_2_O releases Fe^+2^ into the medium, and this occurs at around 130 °C. Therefore, the generation of magnetite in samples F300, S50F, S100F, S150F, and S200F could be due to the high concentration of Fe^+2^ in the medium generated by the rapid solubilization of Fe_3_(PO_4_)_2_·8H_2_O, which in turn would allow the anaerobic oxidation of Fe^+2^ into Fe^+3^ by the water in the medium, as described by Schikorr reaction. On the other hand, the absence of magnetite in the samples S300 and S250F could be explained by the slow and gradual solubilization of most of the Fe_3_(PO_4_)_2_·8H_2_O, which did not cause the medium Fe^+2^ supersaturation. It is worth noting that substantial differences could be observed between the amount of magnetite generated in the F300, S50F, and S100F samples (higher magnetite peaks) against that generated in the S150F and S200F samples (lower magnetite peaks). This lesser amount of magnetite impurities could be mainly explained by the two following factors: firstly, a slow heating rate implemented over the solubilization stage of the Fe_3_(PO_4_)_2_·8H_2_O, (from 130 °C until 150 °C and from 130 °C until 200 °C; respectively) that causes the Fe^+2^ to be generated progressively in the medium, and thus the Schikorr reaction is minimized; secondly, at 150 °C, a certain amount of LiFePO_4_ has already been generated and, consequently, there is a smaller amount of Fe^+2^ in the medium.

Regarding crystallinity (see Figure 6), significant differences can also be observed between the F300 S50F, S100F samples and the S150F, S200F, S250F, S300 samples. Thus, the former samples were less crystalline when compared with the base LiFePO_4_, which could be due to the rapid generation and growth rate of the LiFePO_4_ particles as a consequence of the medium Fe^+2^ supersaturation. Consequently, this rapid particle growth would lead to a poor crystallization, i.e., a poor rearrangement of the atoms in space. On the other hand, the latter samples, which had been subjected to slow heating over 130 °C, presented a substantially improved crystallization. Particularly, the samples from experiment S150F exhibited an optimum crystallization level. In fact, the samples obtained at temperatures over 130 °C did not present any improved crystallinity but rather a poorer one was observed. According to our previous study [34], maintaining the LiFePO_4_ particles at high temperatures (300 °C) for a long time would negatively affect their crystallinity. This could explain why the sample from experiment S150F was the most crystalline, given that the LiFePO_4_ particles that had been formed in the 130 to 150 °C temperature range took just around 8.5 min to reach 300 °C from the moment they had been formed. This means that the length of time that they were subjected to high temperatures was relatively short (see Table 3). The high crystallinity of the S150F sample can be seen in Figure 7, where it is compared against the base sample and a close similarity can be observed.

Therefore, an abrupt change with respect to the generation of magnetite impurities and crystallinity was observed between the S100F and S150F samples that would allow to separate them into two groups as follows: samples F300, S50F, and S100F would fall into the low crystalline particles with extremely high magnetite peaks; while samples S150F, S200F, S250F, and S300 would be classified as highly crystalline particles with low or no magnetite peaks. Figure 7 shows the XRD patterns in more detail, so that the characteristic peak of the base LiFePO_4_ as well as those of S100F, S150F, and S200F samples can be observed. Furthermore, Figure 7 includes the diffractogram of the LaB_6_ sample, which was measured under the same conditions as the base LiFePO_4_ and the S100F, S150F, and S200F samples, so that their crystallite size would be more accurately measured after removing the instrumental broadening. The full widths at half maximum (FWHM) of the peaks (101), (301), and (311) of the synthesized samples were considered to determine the size of the crystallite (*D*) according to Debye-Scherrer equation:(11)$$D=\frac{k\;\lambda }{\beta \;\cos\theta }$$
where λ is the X-ray wavelength in nanometers (nm), β is the full width at half maximum of the peak in radians, θ is the scattering angle in radians, and k is a constant related to crystallite shape, at 0.9 for the Bragg reflections of LiFePO_4_. The instrumental broadening effect on FWHM was removed using Warren’s method on the assumption of a Gaussian peak [43]:(12)$$\beta^{2}=\beta_{sample}^{2}-\beta_{instrumental}^{2}$$
where *β_instrumental_* is referred to LaB_6_ peaks at 21.64°, 30.43°, and 37.35° associated with the planes (101), (301), and (311) of the LiFePO_4_ samples, and correspond to the values of 0.079, 0.075, and 0.067, respectively.

Table 4 shows that the lattice parameters of the LiFePO_4_ particles were similar for all the samples. The Scherrer crystallite size (D) was much smaller than the particle size in every case. The average size of the base LiFePO_4_ sample was 82.5 nm, while the S150F sample was 34.26 nm, the S150F sample was 66.6 nm and the S200F sample was 39.23 nm according to Scherrer’s equation. Therefore, D_base_ LiFePO_4_ > D_S150F_ LiFePO_4_ > DS100F LiFePO_4_ > D_S200F_ LiFePO_4_ due to a further growth of the crystals during the synthesis process. This claim can be confirmed by comparing the following SEM images: base LiFePO_4_ (Figure 3), S100F, S150F, and S200F (Figure 8B–D, respectively).

In addition, the XRD patterns were fitted following the Rietveld method and the orthorhombic space group Pnma. Thus, the amount of magnetite impurities in the S100F, S150F and S200F samples (17.89%, 3.04%, and 2.81% of Fe_3_O_4_, respectively) were quantified and compared against that of the base LiFePO_4_ sample (0.26% of Fe_3_O_4_).

Thereby, the values obtained for the samples studied confirmed the abrupt change presented by the interface of the S100F and S150F samples with regard to magnetite impurity and crystallinity, which made clear that the process requires the use of a reducing agent.

It can be, therefore, be considered that the heating rate within the 130–150 °C temperature range represents a crucial factor if highly crystalline and pure particles are to be obtained while using a smaller amount of reducing agents so that process costs can be kept under control.

#### 3.3.2. Growth Orientation and Morphology of the LiFePO_4_ Particles

In order to investigate the effect from different heating configurations on the morphology and size of the LiFePO_4_ particles, all the samples were subjected to SEM analysis (see Figure 8).

In Figure 8A,B show that the reaction mediums that had not been slowly heated to a temperature of at least 130 °C generated products with widely varied morphology, where particles with fewer edges stand out (S50F, S100F). However, the samples that had been slowly heated over 130 °C (pictures C–F) presented a clear morphology mostly formed by hexagonal microparticles (S150F, S200F, S250F, and S300). In order to contrast and clearly demonstrate the crystals generated with a greater number of facets on the samples showed in Figure 8C–F, the five high peaks corresponding to the planes were selected as follows: (2 0 0), (1 0 1), (1 1 1), (0 2 0), and (3 0 1), and they were compared with the most intense peak corresponding to the (3 1 1) plane (Table 5).

The intensity ratios confirm the increasing trend that appears when the samples are generated by slow heating up to over 130 °C, by which the particles form a greater number of faces. It has, therefore, be demonstrated how the supersaturation of Fe^+2^ in the medium has a remarkable impact.

It should also be noted that that the hexagonal particles in sample S150F (see Figure 8) are evidently thinner, i.e., they have a shorter channel length [10] (see Figure 9). Given that Ceder et al. [44], had already demonstrated that lithium-ion diffusion is several orders of magnitude greater in the [10] direction than in the [1] and [101] directions, and also that M. Saiful Islam et al. [45] proved through simulations, that the energy of Li-ion migration was lower in the [10] direction than in the [1] and [101] directions (Emig [10] = 0.55 eV, Emig [1] = 2.89 eV, and Emig [101] = 3.36 eV), we can affirm that the channel length [10] is a crucial factor to determine particle functionality. The increment in length that experiments the main diffusion channel of lithium ions as the slow heating time is increased (S200F, S250F and S300) may be due to the growth pattern followed by these particles (dissolution–crystallization by Ostwald Ripening, OR) [38]. Even though the particles grow at a greater extent in other directions rather than in the [10] direction of the channel, it is a fact that they also grow in this direction over time. It should also be highlighted how as the heating time increases, size variations are also reduced. This has been corroborated by the results obtained from the different heating configurations applied in the experiments. For example, the particles in sample S150F (as can be seen in Table 3) are formed for 5.5 min under slow heating conditions, and later on, for another 3 min, more particles are formed under fast heating conditions. This difference is less pronounced for example in the S300 sample.

Therefore, it has been demonstrated that heating is a really essential factor at the same level as pH or lithium concentration regarding the generation of an optimal morphology, just as K. Dokko et al. [46], had already reported.

## 4. Conclusions

Our study has demonstrated that the configuration of the heating procedure poses a significant impact on the hydrothermal synthesis of LiFePO_4_ microparticles. Thus, the optimal values to be applied to such heating procedure in order to achieve the desired balance between particle quality and industrial viability could be deemed as the main innovative contribution by our study.

Industrial processes generally aim at the rapid production of particles, so that both resources and operating costs can be reduced. According to the results obtained from the present study, when high heating rates are implemented (F300 samples), the resulting particles present widely varying morphology, poor crystallinity, and considerable amounts of magnetite impurities. On the contrary, the particles that were generated through slow heating exhibited the opposite characteristics, i.e., consistent morphology, good crystallinity, and lower magnetite content, even if their industrial production could present some difficulties associated with the long times that would be required to form the particles (67.5 min in the case of S300 samples).

It has also been confirmed that an abrupt change can be observed in the S100F and S150F samples interfaces, with quite significant variations in their crystallinity as well as in magnetite impurities content (17.89% and 3.04%, respectively). This fact has allowed the separation of the samples into two groups as follows: firstly, the F300, S50F and S100F samples, with low crystalline particles and extremely high magnetite peaks; and secondly the S150F, S200F, S250F, and S300 samples, containing highly crystalline particles and presenting low or no magnetite peaks. Therefore, it has been verified that the samples that were subjected to slow heating rates above the minimum LiFePO_4_ particle formation temperature (130 °C) notably improved their crystallinity and reduced their magnetite content, with sample S150F presenting an optimum crystallinity and morphology.

It can generally be concluded that for the formation of large amounts of LiFePO_4_ particles with good crystallinity levels, it is essential to implement low heating rates within the 130–150 °C temperature range. This would reduce the solubilisation of Fe_3_(PO_4_)_2_·8H_2_O that starts off within that temperature range and causes the medium Fe^+2^ supersaturation and its subsequent oxidization, which in turn results in a poorer crystallization of the LiFePO_4_ particles. Nevertheless, rather long operating times are still required and the use of reducing agents, even if at lesser amounts, would be highly recommended.

In order to reduce operating times and improve industrial viability, the following procedure should be used: extremely fast heating from room temperature to 130 °C, low heating rate from 130 °C to 150 °C, and a final fast heating rate period until the desired final temperature is reached. Thanks to the actual capacity of the current industrial means to implement this fast heating rate in just a few seconds, this procedure should allow the production of optimally crystallized and rather size-consistent particles in just a few minutes. In addition, this process time could be further reduced by adding organic acids to the hydrothermal synthesis, such as J. Ni et al. [47] have already demonstrated.

In an attempt to summarize and clearly display the conclusions reached by this study, the effects from the varying parameters that have been analyzed are presented in Table 6.

## Figures and Tables

**Figure 1 nanomaterials-11-02412-f001:**
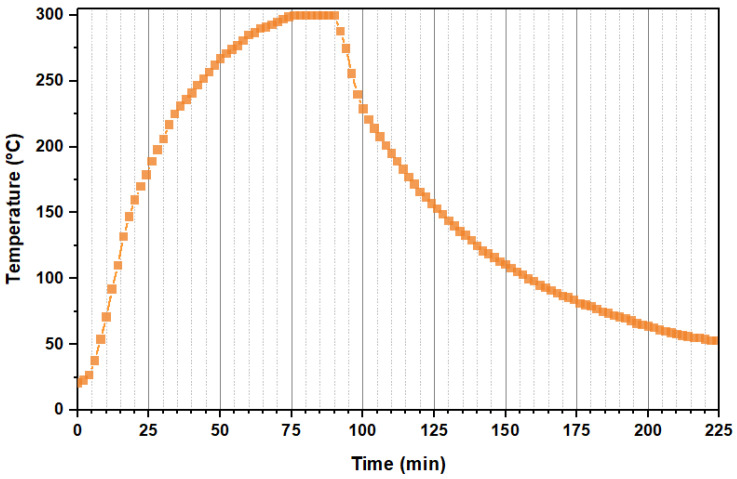
Temperature profile of the reaction medium during the hydrothermal synthesis of the base LiFePO_4._

**Figure 2 nanomaterials-11-02412-f002:**
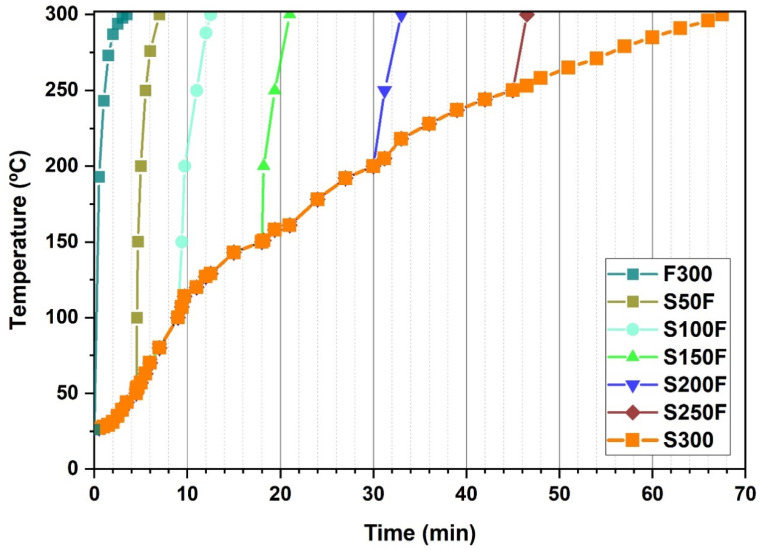
Temperature profile of the reaction medium during the combined slow-fast heating experiments to form LiFePO_4_ microcrystalline particles.

**Figure 3 nanomaterials-11-02412-f003:**
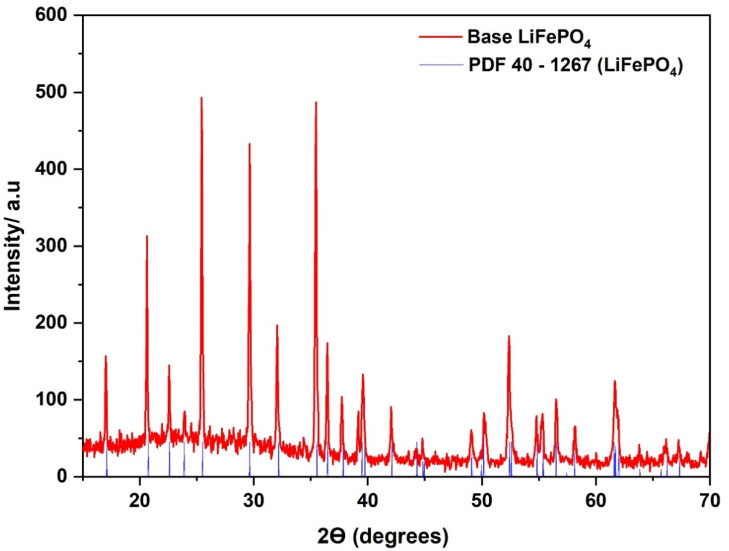
XRD pattern of the synthesized base LiFePO_4_.

**Figure 4 nanomaterials-11-02412-f004:**
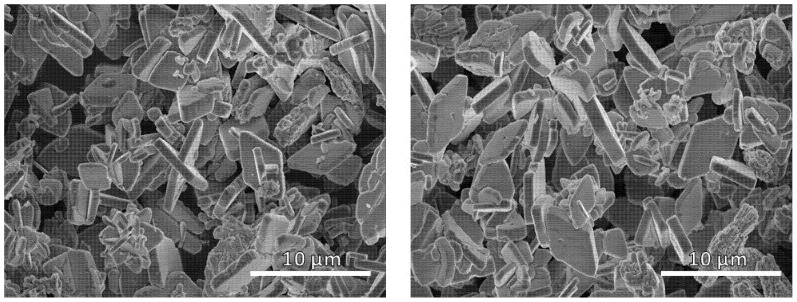
SEM micrographs of the synthesized base LiFePO_4_ obtained from the HS base experiment (two random areas of the sample are displayed).

**Figure 5 nanomaterials-11-02412-f005:**
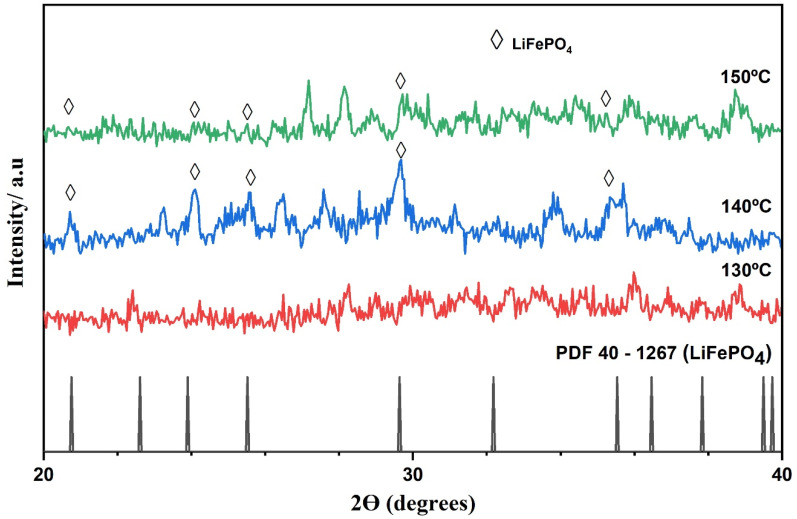
XRD patterns of the LiFePO_4_ particles obtained by hydrothermal synthesis at 130, 140 and 150 °C.

**Figure 6 nanomaterials-11-02412-f006:**
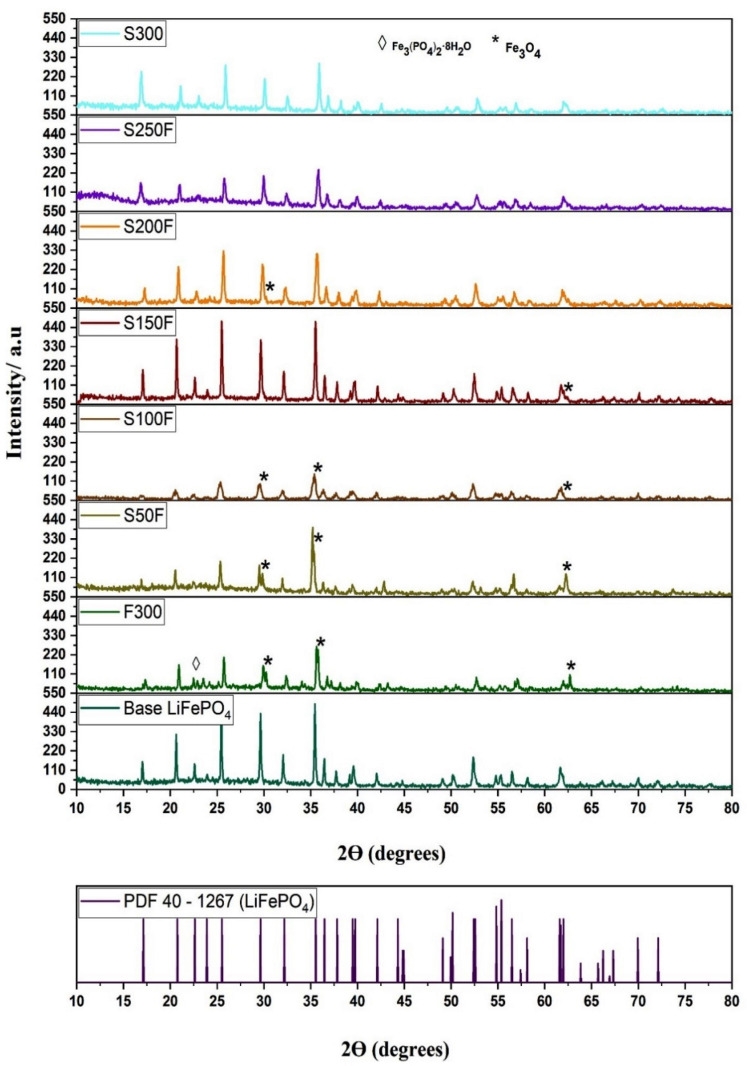
XRD patterns of hydrothermally synthesized LiFePO_4_.

**Figure 7 nanomaterials-11-02412-f007:**
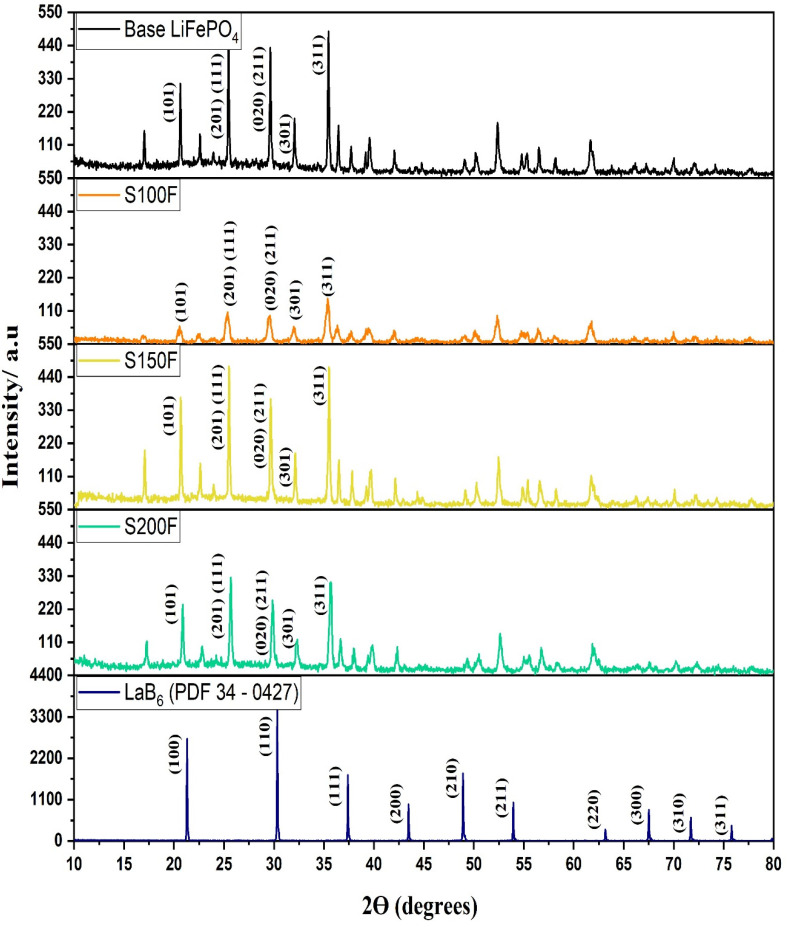
XRD patterns of synthesized Base LiFePO_4_ vs. S150F. The LaB_6_ sample (blue line) was used as the benchmark to determine the instrumental broadening.

**Figure 8 nanomaterials-11-02412-f008:**
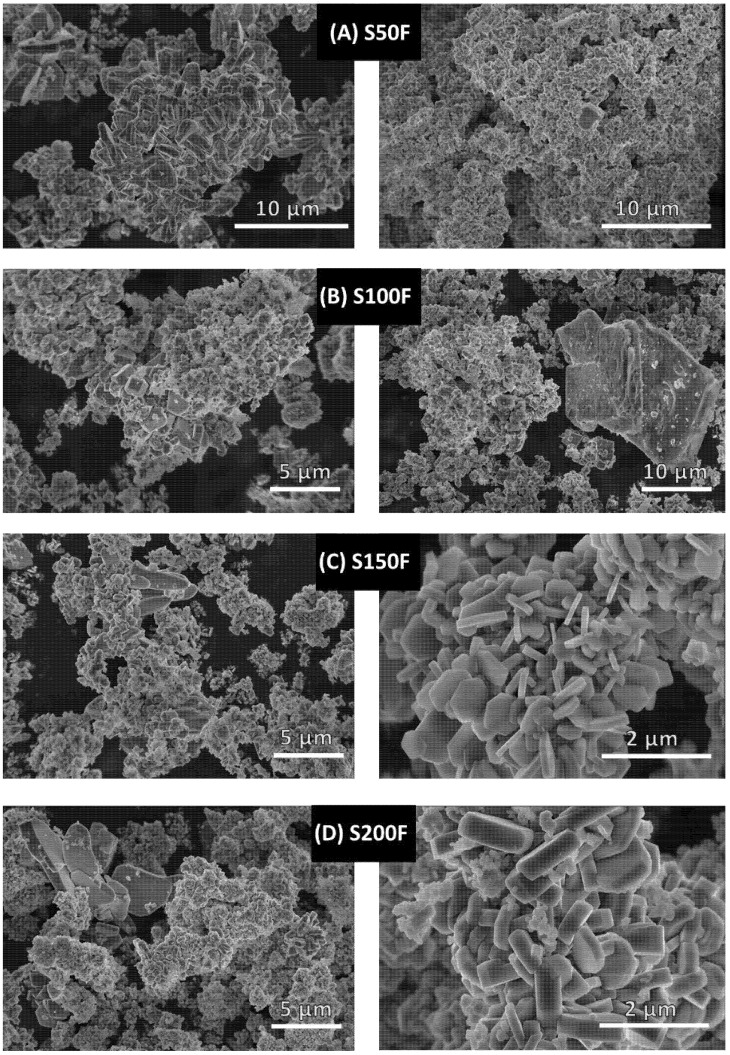
SEM micrographs corresponding to the following experimental samples: (**A**) S50F, (**B**) S100F, (**C**) S150F, (**D**) S200F, (**E**) S250F, (**F**) S300.

**Figure 9 nanomaterials-11-02412-f009:**
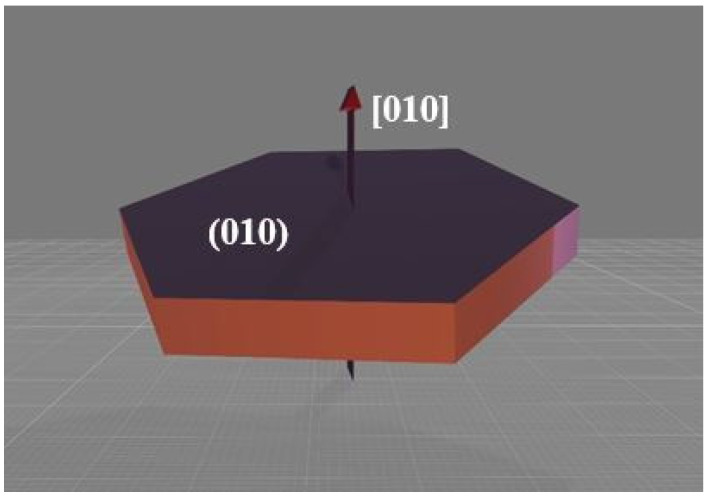
Direction of the particles’ channel [010].

**Table 1 nanomaterials-11-02412-t001:** Nomenclature of the samples and their corresponding experimental conditions. All the samples were heated up to 300 °C and subjected to 110 bars of pressure.

HS Experiments	Slow Heating T_o_ (°C)	Fast Heating T_o_ (°C)	Reaction Time (min)	Total Time Length of the Synthesis Process * (min)
F300	-	300	5	0 + 3.5 + 5 = 8.5
S50F	50	300	5	4.5 + 3 + 5 = 12.5
S100F	100	300	5	9 + 3 + 5 = 17
S150F	150	300	5	18 + 3 + 5 = 26
S200F	200	300	5	30 + 2.5 + 5 = 37.5
S250F	250	300	5	45 + 2 + 5 = 52
S300	300	-	5	67.5 + 0 + 5 = 72.5

* Total time length of the synthesis process: slow heating + fast heating + reaction time.

**Table 2 nanomaterials-11-02412-t002:** Crystalline phases found in the XRD patterns of the synthesized LiFePO_4_ particles.

HS Experiment	Crystalline Phases
F300	LiFePO_4_ + Fe_3_O_4_ + Fe_3_(PO_4_)_2_·8H_2_O
S50F	LiFePO_4_ + Fe_3_O_4_
S100F	LiFePO_4_ + Fe_3_O_4_
S150F	LiFePO_4_ + Fe_3_O_4_
S200F	LiFePO_4_ + Fe_3_O_4_
S250F	LiFePO_4_
S300	LiFePO_4_

**Table 3 nanomaterials-11-02412-t003:** Approximate times to reach operation temperatures.

HS Experiments	T_i_ *	T_s_ *	T_f_ *
S150F	12.5	5.5	3
S200F	12.5	17.5	2.5
S250F	12.5	32.5	2
S300	12.5	55	-

* T_i_: Time elapsed from the beginning of the heating of the reactor until the moment the first crystalline nuclei of LiFePO_4_ are formed. * T_s_: Time elapsed from the moment that the first crystalline nuclei of LiFePO_4_ are formed until fast heating is implemented. * T_f_: Time elapsed from the moment that fast heating is implemented and 300 °C is reached.

**Table 4 nanomaterials-11-02412-t004:** Structural parameters derived from the XRD patterns the synthesized LiFePO_4_ particles and Fe_3_O_4_ impurity content.

Samples	Planes	Peak º (2ϴ)	FWHM	Crystallite Size * (nm)	Average Crystallite Size * (nm)	LiFePO_4_ (%)	Fe_3_O_4_ (%)
*Base* LiFePO_4_	(101)	20.643	0.110	109	82.5	99.74	0.26
	(301)	32.506	0.136	73.5			
	(311)	35.443	0.144	65.0			
*S100F*	(101)	20.949	0.195	45.4	34.26	82.11	17.89
	(301)	29.715	0.320	26.5			
	(311)	35.539	0.278	30.9			
*S150F*	(101)	20.685	0.120	88.2	66.63	96.96	3.04
	(301)	32.133	0.162	57.5			
	(311)	35.515	0.168	54.2			
*S200F*	(101)	20.860	0.173	52.0	39.23	97.19	2.81
	(301)	32.292	0.257	33.2			
	(311)	35.681	0.265	32.5			

* Parameter calculated considering the instrumental broadening.

**Table 5 nanomaterials-11-02412-t005:** Relative intensity ratios of the diffraction peaks in the XRD patterns of the synthesized LiFePO_4_.

Intensity Ratios	F300	S50F	S100F	S150F	S200F	S250F	S300
I (200)/(311)	0.31	0.38	0.22	0.42	0.37	0.69	0.84
I (101)/(311)	0.64	0.59	0.47	0.79	0.76	0.64	0.56
I (111)/(311)	0.82	0.78	0.81	1.01	1.06	0.64	0.56
I (020)/(311)	0.63	0.70	0.71	0.78	0.77	0.85	0.70
I (301)/(311)	0.40	0.41	0.44	0.40	0.38	0.43	0.36

**Table 6 nanomaterials-11-02412-t006:** Summary of effects from the varying parameters considered in this study on the following particle characteristics: purity, crystallinity, morphology, and size.

Samples	Purity and Crystallinity	Morphology and Size	Synthesis Process Time * (min)
F300	→ Low crystallinity → Formation of impurities	Widely variable morphology	8.5
S50F	→ Low crystallinity → Formation of impurities	Widely variable morphology	12
S100F	→ Low crystallinity → Formation of impurities	Widely variable morphology	17.5
S150F	→ Extremely high crystallinity → Reduced impurity formation	→ Hexagonal based particles → Shorter lithium-ion diffusion channel. → Larger particle size distribution	26
S200F	→ Good crystallinity → Reduced impurity formation	→ Hexagonal based particles → Longer lithium-ion diffusion channel. → Decreased particle size distribution	38
S250F	→ Good crystallinity → High purity	→ Hexagonal based particles → Longer lithium-ion diffusion channel. → Decreased particle size distribution	52
S300	→ Good crystallinity → High purity	→ Hexagonal based particles → Longer lithium-ion diffusion channel. → Decreased particle size distribution	67.5

* Synthesis process time: including slow heating + fast heating + reaction time.

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
