# Peer review of "Effect of the Heating Rate to Prevent the Generation of Iron Oxides during the Hydrothermal Synthesis of LiFePO4"

_nanomaterials, 2021, doi:10.3390/nano11092412_

Round 1

Reviewer 1 Report

In this work, the authors applied two heating rates (slow and fast) to the same hydrothermal synthesis process with the main objective of obtaining the purest possible LiFePO4 in the shortest time. The suggestions are given below.

  1. For impurities such as FeOx, XRD cannot detect them because of poor crystallinity but Raman spectroscopy can. The authors can apply Raman technique to characterize the impurities.
  2. For the hydrothermal synthesis of LFP, previous studies have already carried out systematic investigations. Important studies such as J. Mater. Chem. 17 (2007) 4803–4810 and Journal of Power Sources 195 (2010) 2877–2882 are suggested to referred to and discussed.
  3. No electrochemical performances have been reported in this work. This is not usual for a battery material.
  4. Writing is acceptable but proof reading is recommended.

Author Response

We appreciate your comments and review and will proceed to answer all your questions on by one (attached file).

Reviewer 2 Report

In this manuscript, the authors study the effect of the heating rate to prevent the generation of iron oxides during the hydrothermal synthesis of LiFePO4. The results in this manuscript make it interesting and valuable for the researchers. 

Author Response

Thank you very much. We appreciate your comment and review.

Reviewer 3 Report

This work was devoted to investigating the influence of hydrothermal synthesis's heating rates on the purity of the product LiFePO4. Currently, the industrial manufacture technique of LiFePO4 is based on the solid-phase synthesis by using Li2CO3, FePO4, and sucrose as raw materials. This very mature method has many advantages, including negligible waste emission, scalability, and low cost. I do not think that the hydrothermal synthesis can be applied practically in future manufacture, considering the harsh reaction conditions that require high hydrothermal temperature (over 300 °C), high pressure, and high waste-water emission. Therefore, I feel that this work is meaningless, although the authors made quite detailed studies. I do not recommend the publication of this article on Nanomaterials.

Author Response

(The authors gave the same response as above.)

Round 2

Reviewer 1 Report

An improved version.

Author Response

We thank you for yours useful comments and suggestions for improving our manuscript.

Reviewer 3 Report

The authors argued that the hydrothermal synthesis of LiFePO4 may have a promising future, but I still hold to my own opinion, based on the following reasons:

(1) The current industrial technique (i.e., solid-state method) for synthesizing LiFePO4 is very mature and has massive superiority over the hydrothermal synthesis method. The former can afford large-scale, low-cost, continuous manufacturing almost without liquid or solid waste emission, but the latter does not.

(2) The hydrothermal conditions presented by the authors are incredibly harsh, which requires a high temperature of 300 °C, albeit far lower than that of solid-state synthesis. However, this temperature would result in ultrahigh autogenic pressure in a sealed reactor, which is dangerous. In contrast, solid-state synthesis operated under ambient pressure.

(3) I also made a rough estimation. Based on the data in the experimental section, obtaining ~2.5 g LiFePO4 will lead to ~200 mL of waster water, which is entirely unacceptable for industrial implementation.

Thus, I still think this is meaningless research.

Author Response

We appreciate your comments and review.
